# Macroeconomic impact of Ebola outbreaks in Sub-Saharan Africa and potential mitigation of GDP loss with prophylactic Ebola vaccination programs

Laura T. R. Morrison[1][☉]*, Benjamin Anderson[1][☉], Alice Brower[2][‡], Sandra E. Talbird[1][‡], Naomi Buell[1][‡], Pia D. M. MacDonald[1][‡], Laurent Metz[3][‡], Maren Gaudig[3][‡], Valérie Oriol Mathieu[4][‡], Amanda A. Honeycutt[1][‡]

1 RTI International, Research Triangle Park, Triangle Park, North Carolina, United States of America, 2 The Fabulous Co., Durham, North Carolina, United States of America, 3 Johnson & Johnson Global Public Health, New Brunswick, New Jersey, United States of America, 4 Janssen Vaccines & Prevention B.V., Leiden, The Netherlands

☉ These authors contributed equally to this work.
‡ AB, SET, NB, PDMM, LM, MG, VOM and AAH also contributed equally to this work.
* lmorrison@rti.org

**Data Availability Statement:** All data files and original code are available from Zenodo repository, DOI: 10.5281/zenodo.7758492. All data used in the

## Abstract

### Introduction

Decisions about prevention of and response to Ebola outbreaks require an understanding of the macroeconomic implications of these interventions. Prophylactic vaccines hold promise to mitigate the negative economic impacts of infectious disease outbreaks. The objective of this study was to evaluate the relationship between outbreak size and economic impact among countries with recorded Ebola outbreaks and to quantify the hypothetical benefits of prophylactic Ebola vaccination interventions in these outbreaks.

### Methods

The synthetic control method was used to estimate the causal impacts of Ebola outbreaks on per capita gross domestic product (GDP) of five countries in sub-Saharan Africa that have previously experienced Ebola outbreaks between 2000 and 2016, where no vaccines were deployed. Using illustrative assumptions about vaccine coverage, efficacy, and protective immunity, the potential economic benefits of prophylactic Ebola vaccination were estimated using the number of cases in an outbreak as a key indicator.

### Results

The impact of Ebola outbreaks on the macroeconomy of the selected countries led to a decline in GDP of up to 36%, which was greatest in the third year after the onset of each outbreak and increased exponentially with the size of outbreak (i.e., number of reported cases). Over three years, the aggregate loss estimated for Sierra Leone from its 2014–2016 outbreak is estimated at 16.1 billion International$. Prophylactic vaccination could have

analysis are publicly available from the following: • United Nations Human Data Exchange. Ebola Outbreaks Before 2014; n.d. Database: OCHA Services. Available from: https://data.humdata.org/dataset/ebola-outbreaks-before-2014 • The World Bank. Data catalog; n.d. Database: World Development Indicators (WDI). Available from: https://datacatalog.worldbank.org/dataset/world-development-indicators • World Health Organization. Health worker Ebola infections in Guinea, Liberia and Sierra Leone. 2015. Available from: https://www.who.int/csr/resources/publications/ebola/health-worker-infections/en/ • United Nations Human Data Exchange. Number of health-care workers infected with Ebola; n.d. Database: OCHA Services. Available from: https://data.humdata.org/dataset/number-of-health-care-workers-infected-with-edv.

**Funding:** This work was funded by Johnson & Johnson Global Public Health: Johnson and Johnson Services, Inc., One Johnson & Johnson Plaza, New Brunswick, NJ 08933, USAThe funder did not play a role in the study design, conduct, collection, and management but did provide feedback on data analysis and interpretation, as well as manuscript preparation, review, approval and decision to submit.

**Competing interests:** This research was sponsored by Johnson and Johnson Services, Inc., and is related to the development of products licensed to Johnson and Johnson Services, Inc. LTRM, BA, AB, SET, NB, PDMM, and AAH are employees of RTI International which is an independent research organization that received funding from Johnson and Johnson Services, Inc. for the conduct of this study. LM and MG are employees of Johnson & Johnson Global Public Health, and VOM is an employee of Janssen Vaccines & Prevention B.V., which are companies of Johnson & Johnson Services, Inc. This does not alter our adherence to PLOS ONE policies on sharing data and materials.

prevented up to 89% of an outbreak's negative impact on GDP, reducing the outbreak's impact to as little as 1.6% of GDP lost.

## Conclusion

This study supports the case that macroeconomic returns are associated with prophylactic Ebola vaccination. Our findings support recommendations for prophylactic Ebola vaccination as a core component of prevention and response measures for global health security.

## Introduction

The link between infectious disease outbreaks and a weakened economy has perhaps never been as striking in modern times as during the COVID-19 pandemic. However, outbreaks of infectious diseases need not reach pandemic scale to impose substantial negative economic impact.

The world's largest recorded outbreak of Ebola virus disease (EVD) occurred in 2014–2016 in West Africa, with 28,606 reported deaths across Guinea, Liberia, and Sierra Leone [1]. This outbreak disrupted the region's social and economic well-being [2], with an impact on GDP of $2.8–32.6 billion [3] and an overall cost of $53.2 billion, when taking into account costs of mortality, morbidity, and socioeconomic impacts [4]. Many factors contributed to the economic toll of the 2014–2016 outbreak, including decreased demand for goods and services [5], disruptions in international trade [6], increased unemployment, and decreased overall economic activity [7].

The 2002–2004 severe acute respiratory syndrome (SARS) outbreak resulted in 8,403 cases, primarily in Asia [8], and is estimated to have cost between $12.3 billion and $28.4 billion during 2003 in East and Southeast Asia. During the same period, gross domestic product (GDP) was reduced in this region, with the greatest impact observed in China and Hong Kong (estimated GDP reduction of 1.1% and 2.6%, respectively) [9]. A study modeling the potential costs of the economic burden of the Zika virus in six U.S. states estimated that it could cost up to $1.2 billion in medical costs and lost productivity [10]. Moreover, an influenza pandemic is estimated to cost up to $500 billion globally per year [11].

Interventions to prevent and mitigate infectious disease outbreaks, such as vaccination programs, hold the dual promise of reducing both health and economic impacts of these outbreaks [12, 13]. Prophylactic vaccination, in particular, has potential to minimize the economic burden of outbreaks by directly protecting a population from the health impact of disease. Vaccination also allows social and economic activities to continue unhampered by distancing restrictions and fear-induced behaviors that limit production and consumption [14].

Preventive strategies require proactive investment and planning before an outbreak begins, especially for vaccines and other biomedical countermeasures that require time and resources to develop and deliver. Generally, the cost of prevention is lower than that of infectious disease response [15–17] and, specifically, that preventive vaccination is less costly than a reactive response [17–19].

Relatively few studies have estimated the potential for prevention strategies, such as vaccines, to mitigate the negative economic impact of an infectious disease outbreak [20–22]. As leaders make decisions about prevention of and response to Ebola outbreaks, it is important to understand the extent that vaccination may mitigate the negative economic impacts of infectious disease outbreaks. Previous studies provide limited evidence to support these decisions,

as they have primarily evaluated short- and intermediate-term economic impacts of outbreaks, have been based on hypothetical outbreaks, and have not employed approaches to identify causal effects.

The objective of this study was to evaluate the relationship between outbreak size and economic impact among countries with recorded Ebola outbreaks and to quantify potential benefits of prophylactic Ebola vaccination interventions if they had been used during these outbreaks. The economic impact of Ebola outbreaks and of prophylactic Ebola vaccine interventions was estimated for five countries that experienced Ebola outbreaks between 2000 and 2016, during which no vaccines were deployed. This work aims to provide evidence to support calls for vaccine-based Ebola-prevention strategies.

## Methods

The specific Ebola outbreaks and countries included in this study were selected based on the feasibility of estimating causal effects. The outbreaks needed to be of sufficient size (defined as greater than 100 reported cases) to analyze effects on GDP per capita, be sudden, and be unexpected (such that the period before the outbreak was unaffected by anticipation effects or by prior or ongoing Ebola outbreaks). Outbreaks associated or coincident with other ongoing tumultuous events in the country, such as political unrest, were excluded.

Epidemiological and economic data from five countries that had experienced Ebola outbreaks were used: Sierra Leone, Liberia, and Guinea (2014–2016 West Africa outbreak); Uganda (2000–2001 Gulu, Masindi, and Mbarara outbreak); and the Democratic Republic of the Congo (DRC) (2007 Kasaï-Occidental outbreak). To generalize and compare findings across countries, effect size estimates were used to calculate the relative impact of an Ebola outbreak on GDP per capita in percentage terms. Table 1 summarizes key socioeconomic data for each country for the year preceding each outbreak.

Absolute number of cases reported during historical Ebola outbreaks were used as an indicator of the magnitude of the outbreak itself. We used absolute cases instead of population-adjusted cases (e.g., per capita incidence) to avoid understating the risk of Ebola transmission among large populations and the disease dynamics set in motion by the outbreak. GDP is an aggregate measure of economic activity, including consumption, investment, government spending, and net exports (exports minus imports) in a given country, which accounts for the multiple pathways and multisectoral impacts of infectious disease outbreaks [24]. GDP per

**Table 1. Socioeconomic characteristics in year preceding outbreak, by country.**

| Country (year preceding outbreak) | Sierra Leone (2013) | Liberia (2013) | Guinea (2013) | Uganda (1999) | Democratic Republic of the Congo (2006) |
|---|---|---|---|---|---|
| GDP, PPP adjusted (2011$) | $555.21 | $597.38 | $731.56 | $418.56 | $306.52 |
| Access to electricity (%) | 13.5% | 9.8% | 28.8% | 7.5% | 10.4% |
| Current health expenditure (% of GDP) | 11.6% | 8.2% | 3.5% | N/A | 4.5% |
| Infant mortality rate (per 1,000 live births) | 97.7 | 66.9 | 70.5 | 90.3 | 92.8 |
| Secondary school enrollment, female (% gross population) | 36.3% | 41.5% (2011) | 29.0% (2011) | 7.8% (1997) | 25.5% (2007) |
| Percentage of population in poverty at $1.90 per day (2011 PPP) | 16.7% (2011) | 28.1% (2007) | 38.6% (2012) | 28.3% | 63.6% (2004) |

GDP, Gross Domestic Product; PPP: purchasing power parity

Source: World Development Indicators [23]

Note: Values in table are drawn for year preceding outbreak examined in this study. If data was unavailable, the nearest year of recorded data preceding the outbreak was included and if data preceding the outbreak was unavailable, N/A was recorded.

capita at purchasing power parity (PPP) was employed as a measure of national economic well-being and living standards because it is widely understood and suitable for comparison across countries. Further, GDP accounts for economic change at the population level, where the majority of vaccines' benefits are expected [14]. Adjusting GDP for PPP considers the varying costs of goods and services over time and across countries.

The study was conducted in three steps: firstly, the synthetic control method (SCM) was used to identify the causal effects of five past Ebola outbreaks on per capita GDP at PPP. Secondly, disease incidence was calculated using illustrative vaccine deployment scenarios to estimate upper and lower bounds of the potential cases averted by prophylactic Ebola vaccination. Thirdly, the output from the prior two steps was used to predict the economic benefit of reduced Ebola cases in the context of a prophylactic vaccination program.

## Estimating causal effects of Ebola outbreaks using synthetic control

Using historical outbreaks to analyze the effects of Ebola outbreaks on GDP per capita poses challenges to using classical methods of causal inference, which makes the SCM best suited to this context: one, sizable Ebola outbreaks have struck only a handful of countries, limiting the number of Ebola-affected countries and objectively similar unaffected countries to study; two, no single country or simple average of countries offers a reliable comparison group, or counterfactual, against which to compare the economic outcomes for the Ebola-affected countries because each Ebola-affected country is different from possible comparison countries in terms of GDP per capita trends and other relevant characteristics.

The SCM generates causal effect estimates using aggregate data in settings where there may be only a single treated unit and a small number of comparison units by developing a counterfactual based on a weighted combination of comparison units (the synthetic control) [25]. The weights are determined by an algorithm to minimize the difference between the pre-outbreak characteristics of the Ebola-affected country and the synthetic control. These pre-outbreak characteristics include trends in and predictors of GDP per capita itself and predictors of GDP per capita. The result is a synthetic control that closely resembles the Ebola-affected country for the period before the outbreak (S1 Table).

When using the synthetic control as a counterfactual, the effect of the Ebola outbreak is estimated as the difference between the Ebola-affected country's observed outcomes and the synthetic control in the post-outbreak period. The SCM has been applied to evaluate the impact of interventions on aggregate outcomes, such as GDP [26], and has increasingly been used to estimate causal effects in the fields of public health and epidemiology [27–32]. The similarity of the synthetic control to the Ebola-affected country in the pre-outbreak period is an indication that the synthetic control is effectively accounting for observed and unobserved time-varying factors that affect GDP. The effect of the Ebola outbreak is inferred by the difference between the observed GDP per capita in the Ebola-affected country and its synthetic control in the post-outbreak period. Thus, the synthetic control offers a data-driven counterfactual from which to infer the effect of the Ebola outbreak on GDP which can be tested to assess its reliability and robustness.

This analysis examines GDP per capita at PPP that was lost due to the outbreak accrued in the first three years after the outbreak began, as this timeframe takes into account the growing magnitude of the effects on GDP per capita over time. Absolute values of GDP per capita (U.S. international dollars [Int'l$]) were converted to a percentage of counterfactual GDP per capita, which can be interpreted in the results as the percentage impact on living standards caused by the outbreak.

The predictor variables we used to estimate the synthetic controls have frequently been used in studies employing the SCM to analyze effects on GDP per capita: trade openness,

Polity scores, human capital index, share of GDP from industry, urban population share, share of the population using the internet, and pre-trends of GDP per capita [33–35]. By construction, the synthetic control algorithm develops a set of weights that, when applied to the set of comparison countries (the donor pool), resembles the Ebola-affected country in terms of GDP per capita and in terms of the predictors of GDP per capita. Data were drawn from the Penn World Table and the World Development Indicators [23, 36]. The synthetic control analyses were conducted in Stata using the synth_runner package [37].

The donor pool was determined by quantitative and qualitative criteria aimed at maximizing the comparability of the countries with the Ebola-affected country (for an Ebola outbreak). The included Ebola-affected countries are in sub-Saharan Africa and were among the poorest countries in the world in terms of GDP per capita at PPP. Thus, the donor pool was limited to sub-Saharan African countries within the bottom decile of countries in the world in terms of GDP per capita at PPP in the outbreak year. Any countries bordering the Ebola-affected countries that satisfied these criteria were retained in the donor pool because their inclusion generated conservative estimates in the case that a neighbor indeed suffered from spillover effects from the outbreaks.

Falsification or "placebo" tests and robustness checks were used to assess the reliability of the effect estimates [26]. These entailed estimating synthetic controls for each country in the donor pool as if it had been affected by an Ebola outbreak. This yielded a distribution of placebo effects, which were compared with the effects estimated for the Ebola-affected country. Specifically, differences between the root mean square prediction errors (RMSPEs) estimated for the Ebola-affected country and those of the donor pool were calculated. If an effect is present and the synthetic counterfactual is valid, the RMSPEs in the post-outbreak period should be large relative to the RMSPEs in the pre-outbreak period for the Ebola-affected country. The distribution of post-/pre-RMSPE ratios were also assessed, with the expectation that the RMSPE ratio for the Ebola-affected country would be notably greater than that of most of its donors produced by the placebo tests (S1 File).

To identify further systematic differences between the Ebola-affected country and its donor pool, magnitudes of the effects with those produced by placebo testing were compared. For each year, the proportion of placebo effects that were at least as large as the effect estimated for the country of interest were calculated. These proportions were based on absolute magnitudes and did not consider the direction of the effects. Because the validity of effect estimates depends on the quality of the synthetic control's match in the pre-outbreak period, the standardized placebo effect proportions were also calculated (S3 Table) by dividing the placebo effect proportions by the pre-RMSPEs.

Three types of sensitivity analyses were conducted in order to explore the robustness of these estimates to three fundamentally different alternatives about how to implement the synthetic control analysis. Firstly, the composition of the donor pools for each Ebola-affected country were varied by omitting one donor and generating an alternative synthetic control for comparison iteratively. Secondly, the set of predictor variables used to estimate the synthetic control for each Ebola-affected country were varied by using nine alternative combinations of variables to produce nine alternative synthetic controls. Thirdly, the length of the pre-outbreak period was modified, which can lead to a different set of weights applied to the donor countries and again generate an alternative synthetic control.

## Estimating the reduction in cases from Ebola vaccine deployment

To estimate the impact of prophylactic vaccination strategies on EVD incidence, assumptions were made about vaccine coverage and vaccine efficacy and applied to Ebola case data for each

**Table 2. Vaccination scenarios using the disease incidence calculator.**

| | Scenario 1: Low Coverage/Low Efficacy | Scenario 2: High Coverage/High Efficacy |
|---|---|---|
| Vaccine efficacy (reduced risk of being infected), % | 60 | 90 |
| Vaccinated population who received a second dose, % | 80 | |
| Vaccinated population, % | | |
| High-risk population | 30 | 60 |
| General population | 5 | 10 |

Source: Authors' assumptions.

of the five Ebola-affected countries included in the study. Two illustrative vaccination scenarios were assumed (Table 2): low coverage with low efficacy (i.e., providing a lower bound on cases averted) and high coverage with high efficacy (i.e., providing an upper bound). The assumptions noted in Table 2 were made for each scenario about hypothetical vaccine coverage and vaccine efficacy and applied to Ebola case data. Baseline numbers of probable and confirmed cases in each country were allocated between high-risk and non-high-risk subpopulations, reflecting the disproportionately high risk faced by specific groups [38]. The high-risk subpopulation included all health care workers (doctors, nurses, midwives, and others employed in the hospital and community, including pharmacists, hygiene personnel, laboratory personnel, traditional medicine doctors, and community health workers), frontline health workers, members of the armed forces, and transportation workers (S2 Table). Reported EVD incidence rates among health care workers were applied to the total high-risk subpopulation in each country to estimate the number of cases in this subpopulation (Table 3). The remaining cases, as reported in surveillance data, were assumed to occur in the non-high-risk subpopulation (i.e., general population). Given the lack of relevant economic impact identified in the synthetic control estimation for Guinea (explained in more detail in the Results section), Guinea was excluded as an outlier for the steps of our analysis following the synthetic control estimation. Therefore, cases and estimated cases averted are shown for the remaining four countries included in the analysis.

To determine the potential impact of preventive vaccination on the number of Ebola cases during an outbreak, estimates from a published modeling analysis of the impact of prophylactic vaccination strategies on Ebola virus transmission in the DRC were applied [40], which showed a 62% reduction in cases in a comparable low coverage/low efficacy scenario and a 91% reduction in a comparable high coverage/high efficacy scenario (a detailed description is included in the supporting information).

## Predicting the economic benefits of the prophylactic vaccination

Evidence generated from the synthetic control analyses and from the estimated cases averted from implementation of prophylactic vaccination were used as the basis for predicting the economic effects of the Ebola vaccine. The synthetic control results offer empirical grounding for the relationship between outbreak size (i.e., number of reported cases) and impact on GDP per capita.

The functional form that best fits the evidence pertaining to the relationship between Ebola cases and impact on GDP per capita is exponential, given by

$$f(c) = 2.4625 + e^{2.4638E-04c}, \tag{1}$$

**Table 3. Data inputs and assumptions.**

| Country | Baseline Cases | | | Scenario 1: Low Coverage/Low Efficacy | | | Scenario 2: High Coverage/High Efficacy | | |
|---|---|---|---|---|---|---|---|---|---|
| | All | High-Risk Pop. | Non-High-Risk Pop. | All | High-Risk Pop. | Non-High-Risk Pop. | All | High-Risk Pop. | Non-High-Risk Pop. |
| | Total Cases | Total Cases | Total Cases | Cases Averted | Cases Averted | Cases Averted | Cases Averted | Cases Averted | Cases Averted |
| Sierra Leone | 14,122 | 1,707 | 12,415 | 8,756 | 1,058 | 7,697 | 12,851 | 1,553 | 11,298 |
| Liberia | 10,675 | 2,323 | 8,352 | 6,619 | 1,441 | 5,178 | 9,714 | 2,114 | 7,600 |
| Uganda | 425 | 71 | 354 | 264 | 44 | 220 | 387 | 64 | 322 |
| DRC | 264 | 33 | 231 | 164 | 20 | 143 | 240 | 30 | 210 |

DRC, Democratic Republic of the Congo; Pop., population.

Sources: Total cases: United Nations Humanitarian Data Exchange (2019) [39]; High-risk population cases: United Nations Humanitarian Data Exchange (2015) [1]; Percentage of cases averted: Potluri et al (2020) [40, Technical Report, Tables 14 and 24]. Values drawn from this report were calibrated to the 2018 DRC outbreak and assumed no reduction in infectiousness and case fatality rates from vaccination; see S4 Table for details.

Note: Cases by risk groups were calculated by authors using number of cases and sizes of the high-risk and non-high-risk populations in each year of the outbreak; see S2 Table for underlying data. All values in the table were rounded to the nearest whole case; this rounding accounts for differences in total cases averted and population-specific cases averted.

where $c$ is the number of cases, and $f(c)$ is the negative impact accrued by the third year post-outbreak as a percentage of the counterfactual GDP per capita in that year.

## Incorporating reduction in cases to calculate economic impact of vaccination

We predicted the impact of prophylactic vaccine interventions on GDP per capita using the estimates of cases averted under the illustrative Ebola vaccine intervention scenarios described previously. The estimates of EVD cases averted due to vaccination as inputs to the relationship described by Eq 1 were used to calculate the change in predicted GDP per capita impact due to prophylactic vaccine intervention. The function predicts a change in the outbreak's impact on GDP per capita associated with moving from the actual total cases associated with an Ebola outbreak, $c_0$, to a smaller number of cases characterized by the vaccine at the low coverage/low efficacy scenario, $c^L$, or the high coverage/high efficacy scenario, $c^H$. The difference between $f(c_0)$ and $f(c^L)$, and $f(c_0)$ and $f(c^H)$, yields the predicted economic benefit of the low and high prophylactic vaccination scenarios, respectively, defined in terms of the percentage loss of GDP per capita that is *mitigated*.

## Results

### The effect of Ebola outbreaks on GDP per capita in Ebola-affected countries

Estimates from the synthetic control analysis indicate that past Ebola outbreaks had a negative impact on GDP per capita, causing declines in living standards of between 4.2% and 35.6% by the third year after the start of the outbreak, when impacts were most pronounced. Comparing observed GDP per capita of the five countries included in the study against each country's synthetic control illustrates the estimated effect of the Ebola outbreak (Fig 1). The observed GDP values and synthetic controls are closely matched across the pre-outbreak period, particularly immediately before the EVD outbreaks, and diverge after the outbreaks begin. For all countries, apart from Guinea, the effect size was greatest in the third year after the start of an Ebola outbreak and increased exponentially with the number of EVD cases in an outbreak.

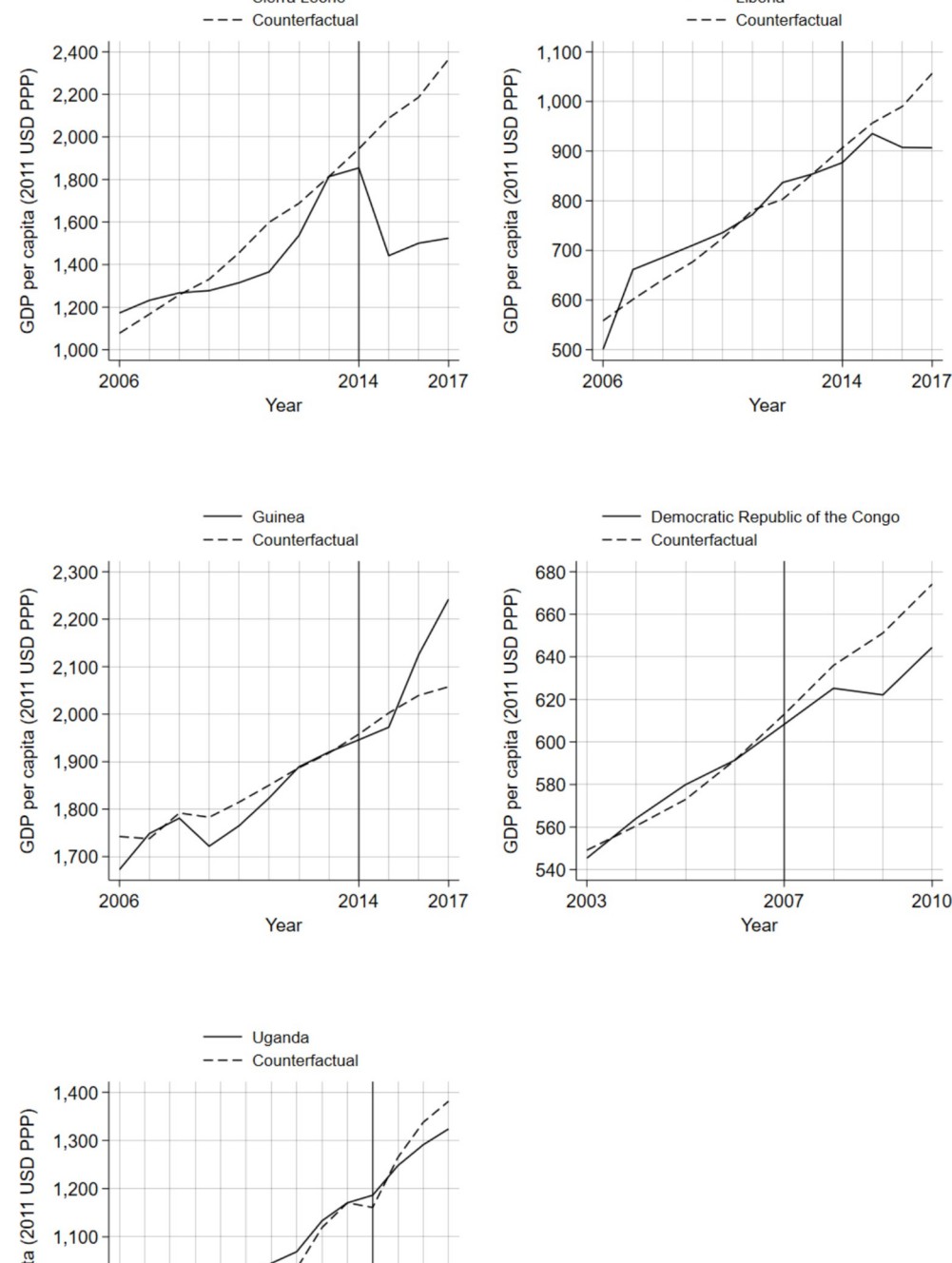

**Fig 1. GDP per capita in Ebola-affected countries and synthetic controls.** GDP, gross domestic product; USD, U.S. dollars. The start of the Ebola outbreak for each country is indicated by the vertical line.

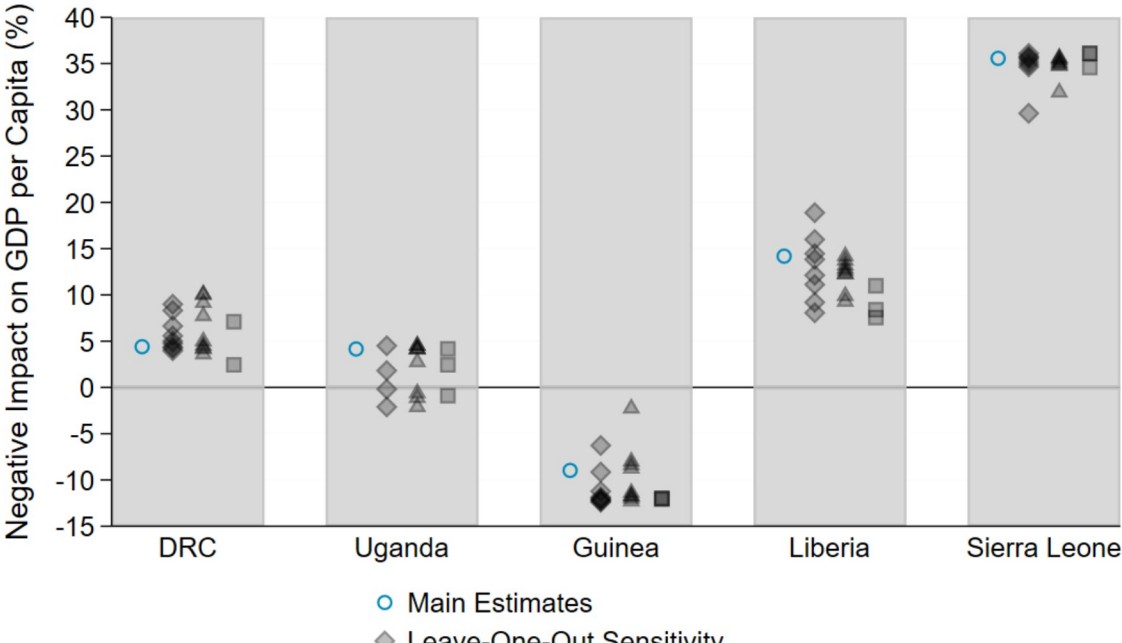

**Fig 2. The cumulative effect on GDP per capita accrued by year 3 post-EVD outbreak: Main estimates versus sets of estimates generated from three types of robustness tests.** DRC, Democratic Republic of the Congo; EVD, Ebola virus disease; GDP, gross domestic product.

Countries with the highest number of EVD cases, Sierra Leone and Liberia, experienced the largest negative economic impacts on GDP per capita, in absolute terms and as a percentage decline in standard of living. By the third year after the outbreaks began, living standards were 35.6% lower in Sierra Leone and 14.2% lower in Liberia compared to the synthetic control. In Uganda and the DRC, both of which experienced relatively small outbreaks, the effects were modest albeit negative. Guinea was anomalous as its actual GDP per capita surpassed its synthetic control over the 3-year post-outbreak period and was therefore not included in subsequent analyses.

Fig 2 plots the percentage impact on GDP per capita produced by each sensitivity test, along with the results from the main specification. The results produced by all three types of robustness tests are consistent with the synthetic control estimates for the Ebola-affected countries in Fig 1, indicating that varying key aspects of the specification does not substantially change the primary results. The consistency of the estimates supports our primary results and provides an indication of the credible ranges for the effects.

## Averted economic loss from vaccination

Fig 3 plots the exponential relationship between Ebola cases and negative impact on GDP per capita by the third year after outbreaks began (i.e., aggregated over years 1–3 post-outbreak). The function fits the impact estimates for the countries reasonably well, considering that heterogeneous effects from one context to the next are probable, and suggests that the number of cases in an Ebola outbreak is a reasonable predictor of that outbreak's effect on GDP per capita in the country where it occurs.

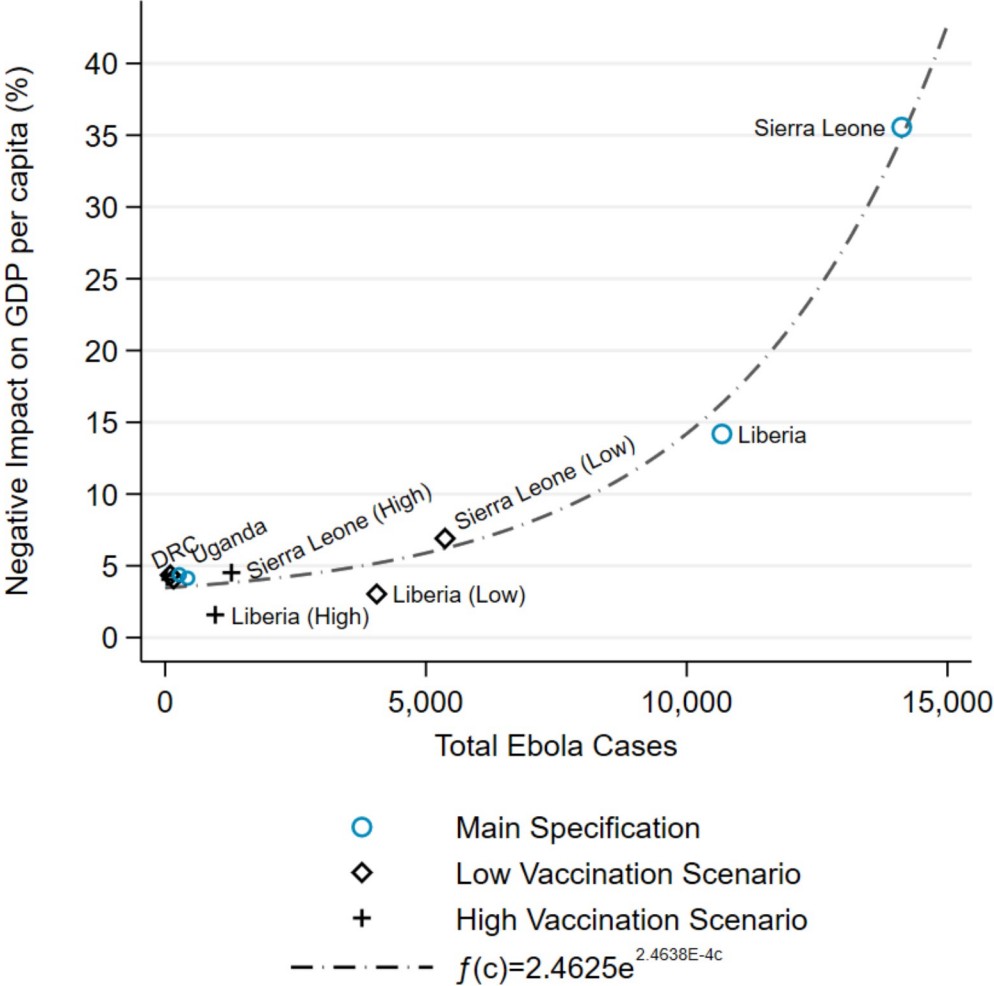

**Fig 3. Function describing the relationship between Ebola cases and GDP impact.** DRC, Democratic Republic of the Congo; GDP, gross domestic product.

Table 4 summarizes the predicted economic benefits for each country from averted cases under each vaccination scenario. Benefits, or averted costs from the outbreak, due to the prophylactic vaccine interventions are economically significant and grow in exponential proportion to the number of cases. The marginal economic benefits of averting a case are exponentially greater in countries with larger outbreaks due to the convexity of the relationship between Ebola cases and economic impact.

The impact on GDP by the third year after the start of the Ebola outbreak was the greatest in Sierra Leone. In the absence of vaccination, a 35.6% loss of GDP was estimated, which could have been reduced to 6.9% or 4.5% loss of GDP in the low- or high-vaccination scenarios, respectively, representing a mitigation of 80.6% to 87.3% of the reduction in GDP. In Liberia, where Ebola had the second greatest impact and caused a 14.2% loss of GDP, it was predicted that the low- and high-vaccination scenarios could have mitigated 78.6% to 88.8% of that reduction, respectively.

This study predicts that the losses caused by the largest outbreak (in Sierra Leone) could have been reduced from 35.6% to 4.5% loss in GDP in the high coverage/high efficacy prophylactic vaccination scenario. Translating the estimated effects to aggregate monetary values for

**Table 4. Estimated economic benefits of prophylactic Ebola vaccine by year 3 post-outbreak, by country.**

| Country | Outbreak Year | Observed | | | Estimated Impact from Hypothetical Vaccination Intervention | | | | | | | |
|---|---|---|---|---|---|---|---|---|---|---|---|---|
| | | | | | Averted Cases | | Total GDP Loss Mitigated by Vaccine (million $, aggregate GDP loss over three years)[c] | | Share of GDP Loss from Outbreak Mitigated by Vaccine (%)[d] | | Remaining GDP Loss (%)[e] | |
| | | Total Reported Ebola Cases | GDP per Capita Loss (%) | GDP per Capita Loss ($) | Low[a] | High[b] | Low[a] | High[b] | Low[a] | High[b] | Low[a] | High[b] |
| Sierra Leone | 2014 | 14,122 | 35.56 | 841.20 | 8,756 | 12,851 | 5,122.91 | 5,548.75 | 80.59 | 87.28 | 6.90 | 4.52 |
| Liberia | 2014 | 10,675 | 14.18 | 149.87 | 6,619 | 9,714 | 557.43 | 629.86 | 78.60 | 88.82 | 3.03 | 1.59 |
| Uganda | 2000 | 425 | 4.15 | 57.38 | 264 | 387 | 25.71 | 37.13 | 1.68 | 2.43 | 4.08 | 4.05 |
| DRC | 2007 | 264 | 4.40 | 29.66 | 164 | 240 | 18.39 | 26.66 | 0.96 | 1.39 | 4.36 | 4.34 |

DRC, Democratic Republic of the Congo; GDP, gross domestic product (2011 U.S. dollars purchasing power parity).

[a] Low coverage/efficacy: 60% efficacy, 30% coverage of high-risk population plus 5% coverage of non-high-risk population.

[b] High coverage/efficacy: 90% efficacy, 60% coverage of high-risk population plus 10% coverage of non-high-risk population.

[c] Total GDP Loss Mitigated by Vaccine represents to the avoided reduction in GDP that is estimated in each scenario as a result of the vaccine. This estimate aggregates the total loss in years one, two, and three following the outbreak.

[d] Share of GDP Impact Mitigated by Vaccine represents the share of GDP loss due to the outbreak that would be mitigated by the vaccine, specifically, in each hypothetical scenario.

[e] Remaining GDP Loss represents the share of total GDP that would still be impacted in each scenario even when the vaccine was deployed (i.e., impact to GDP that is not influenced by the presence of a vaccine.)

the two largest outbreaks, in Sierra Leone and Liberia, results in losses of 841 Int'l$ and 150 Int'l$ per person, respectively, which amounts to losses of about 6.4 billion Int'l$ and 0.7 billion Int'l$. Considering that the outbreaks also caused losses in the first and second years after the outbreak, the aggregate impact between 2015 and 2017 for Sierra Leone and Liberia yields total cumulative estimated losses of 16.1 billion Int'l$ and 1.2 billion Int'l$, respectively.

Tests to illustrate the sensitivity of economic effect estimates to the herd immunity factor used are provided in supporting information (S4 Table).

## Discussion

Deployment of a prophylactic Ebola vaccine could significantly reduce the sizable economic losses and notable impact on living standards caused by Ebola outbreaks, which can have significant ripple effects on socio-economic factors and well-being. These impacts increased exponentially with the outbreak's size (i.e., number of cases), which suggests that in the absence of effective prevention measures, these negative impacts of an Ebola outbreak could grow quickly. Compared with other studies examining the impact of Ebola on macroeconomic outcomes, particularly those focused on the 2014 outbreak in West Africa, these findings generate comparable estimates taking into account that other studies consider a limited set of economic impacts and rely on different methodologies to estimate impact.

This study is the first to shed new light on the macroeconomic implications of vaccination as a prevention strategy for Ebola. Further, this approach provides insights into the relationship between macroeconomic outcomes and the total number of Ebola cases experienced in an outbreak by identifying these effects for multiple individual Ebola outbreaks. This study provides further information on the impact of Ebola outbreaks on GDP of countries where outbreaks occur [3, 4], using quasi-experimental methods to identify these effects.

This study demonstrates how the SCM can be applied to identify the causal impact of an Ebola outbreak on GDP per capita, and the SCM has many advantages in this context because it provides an approach to creating a credible counterfactual from a combination of multiple countries for cases where few affected units exist. This maximizes the comparability of the synthetic control to the country of interest and produces a counterfactual that is less likely to be influenced by extraneous factors than a single comparison country alone. As the availability and uptake of prophylactic Ebola vaccination increases, the SCM may also help to identify causal effects of vaccines on economic outcomes directly and to assess them against complementary and substitute preventive measures.

Interpretation of these results should also consider several limitations of this study. Ebola outbreaks can affect the economy of a country in many ways, and relying on an aggregate measure, such as GDP, may mask the underlying dynamics within the economy. GDP is a composite measure whose subcomponents may not always move in the same direction. Thus, changes in GDP reflect the net change in its components: consumption, investment, government spending, and net exports, each of which could be impacted positively or negatively by an infectious disease outbreak [12]. For instance, the effect on GDP could be masked even if an outbreak decreases household consumption and business investment, provided that these decreases are countered by increases in government spending and foreign aid. GDP is also prone to measurement challenges that can undermeasure or miss economic activity in regions, sectors, or populations for which economic data are not collected. In low- and middle-income countries especially, GDP may be systematically undermeasured in rural areas and instead reflect urban areas where larger firms operate. Aggregate measures can also obscure the social dimensions of Ebola's impact, such as impacts on vulnerable populations and disruption to health care and education [4, 12].

To implement the SCM approach, several country-specific limitations had to be addressed. The 2007 DRC outbreak was included because the pre-outbreak period was free of lagged effects from previous outbreaks and economic turmoil, unlike other outbreaks that have occurred since 2014 in the DRC. The length of the pre-outbreak period in the DRC was also reduced compared with other countries in the study to avoid the influence of these confounding factors. Still, the synthetic control generated a relatively strong counterfactual by closely matching the observed GDP per capita before the outbreak. Additionally, the Uganda outbreak was limited by data availability and constraints on the number of comparison countries available to include in the donor pool. Conflicts and political turmoil also ruled out several countries from Uganda's donor pool, resulting in fewer donors to construct the synthetic control and a less precisely matched synthetic control during the pre-outbreak period. The synthetic control estimates for Uganda may be conservative because the observed GDP per capita of Uganda exceeded its synthetic control before the outbreak and abruptly reversed this after the outbreak.

The outbreak in Guinea led to a counterintuitive result; its observed GDP per capita exceeded the synthetic control, which may be caused by two potentially confounding, coinciding events that did not affect other countries impacted by the 2014 West Africa outbreak. While Sierra Leone and Liberia also received elevated levels of foreign aid during the 2014–2016 Ebola outbreak, Guinea received a debt cancellation of $2.1 billion (two-thirds of its total debt) [41] and a change in a 5-year suspension of aid. [42] These events were both positive for Guinea's economy and unrelated to the Ebola outbreak that would shortly follow, thus their effects could not be disentangled from the effects of the outbreak [43, 44].

In estimating reduced disease incidence directly from prophylactic vaccination and indirectly from herd immunity, we used factors that were not country- or outbreak-specific, yet permitted simplification and application of comparable assumptions around vaccine efficacy

and coverage, and population-level herd immunity. Vaccine efficacy and coverage assumptions that spanned a range of possible outcomes were applied to represent an illustrative upper and lower bound, whereas the herd immunity factor was derived from an Ebola transmission model developed for the DRC [40]. Models that capture country- and outbreak-specific disease transmission dynamics would allow for more precise estimates of the number of cases averted for each country. The simplified approach adopted in the current analysis may not accurately reflect the potential vaccine impact in different Ebola outbreaks.

## Conclusion

These findings support recommendations for prophylactic vaccination as a core component of global prevention efforts and is useful for national public health leaders and multilateral organizations as they prepare for future Ebola outbreaks [45].

## Supporting information

**S1 Table. Pre-outbreak GDP per capita of Ebola-affected countries and prediction error: Synthetic control versus the average of donor countries and the two most similar countries.** DRC, democratic republic of the Congo; GDP, gross domestic product.
(DOCX)

**S2 Table. High-risk population as a fraction of total population (per 1,000), by country.** DRC, democratic republic of the Congo.[a] Includes health care practitioners. Source: (1) World Health Organization (2018); (2) Authors' assumption; (3) World Bank (2020); (4) UNICEF (2020) and Barber and van der Weijde (2019).
(DOCX)

**S3 Table. GDP per capita in Ebola-affected countries and synthetic controls.** DRC, Democratic Republic of the Congo; GDP, gross domestic product; RMSPE, root mean squared prediction error; SC, synthetic control. S3 Table reports the GDP per capita values for the Ebola-affected countries and the synthetic controls, as well as the effects in both absolute terms and percentages. Because levels of GDP per capita vary considerably across countries, the effect sizes reported as a percentage are more useful for cross-country comparisons. The placebo effect proportions associated with the effects estimated for each year and the RMSPE ratio proportions indicate that the effects are unlikely due to chance. This is particularly true for Sierra Leone and Liberia, which suffered the highest numbers of Ebola virus disease cases, and for the effects identified for the third year after each outbreak began for each country because of the apparent persistent and lagged effects.
(DOCX)

**S4 Table. Sensitivity of herd immunity factor on change in GDP per capita effect estimates.**
(DOCX)

**S1 Fig. Distribution of synthetic control estimates for Ebola-affected countries versus placebo treatment applied to control countries.** S1 Fig plots the prediction errors in the pre-outbreak period and effects in the post-outbreak period for the Ebola-affected country. The prediction errors and the placebo effects are plotted for the comparison countries. The outbreak is represented by the vertical line. The ratio of the effect sizes in the post-outbreak period to the size of the prediction errors in the pre-outbreak period were generally much greater for the Ebola-affected countries than for the donor countries that received the placebo treatment. The effect sizes are generally larger than the placebo effects as well.
(TIFF)

**S2 Fig. Main synthetic control estimates versus synthetic controls generated by three types of robustness testing.** S2 Fig graphs the actual GDP per capita of the Ebola-affected countries against the full time series of the alternative specifications alongside the main synthetic control specification. Alternative predictor variables that were added or substituted include capital stock, population density, inflation, electrification, and government expenditure on health care.
(TIFF)

**S1 File. The RMSPE ratio.**
(DOCX)

## Acknowledgments

The authors would like to acknowledge Thierry Van Effelterre (Janssen), Trevor Bacon (Janssen), Ruxandra Draghia Akli (Janssen) and Alex Turner (RTI International) for contributions to the review of this manuscript.

## Author Contributions

**Conceptualization:** Laura T. R. Morrison, Benjamin Anderson, Alice Brower, Pia D. M. MacDonald, Laurent Metz, Maren Gaudig, Valérie Oriol Mathieu.

**Data curation:** Laura T. R. Morrison, Benjamin Anderson, Alice Brower, Naomi Buell.

**Formal analysis:** Laura T. R. Morrison, Benjamin Anderson, Alice Brower, Naomi Buell, Amanda A. Honeycutt.

**Funding acquisition:** Laura T. R. Morrison, Sandra E. Talbird, Pia D. M. MacDonald.

**Investigation:** Laura T. R. Morrison, Benjamin Anderson, Alice Brower, Amanda A. Honeycutt.

**Methodology:** Laura T. R. Morrison, Benjamin Anderson, Amanda A. Honeycutt.

**Project administration:** Laura T. R. Morrison, Sandra E. Talbird.

**Resources:** Laura T. R. Morrison, Sandra E. Talbird.

**Software:** Sandra E. Talbird.

**Supervision:** Laura T. R. Morrison, Benjamin Anderson, Sandra E. Talbird, Pia D. M. MacDonald, Amanda A. Honeycutt.

**Validation:** Laura T. R. Morrison, Benjamin Anderson, Alice Brower.

**Visualization:** Laura T. R. Morrison, Benjamin Anderson, Alice Brower.

**Writing – original draft:** Laura T. R. Morrison, Benjamin Anderson, Amanda A. Honeycutt.

**Writing – review & editing:** Laura T. R. Morrison, Benjamin Anderson, Alice Brower, Sandra E. Talbird, Naomi Buell, Pia D. M. MacDonald, Laurent Metz, Maren Gaudig, Valérie Oriol Mathieu, Amanda A. Honeycutt.

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
