## [Decision Letter · Decision Letter 0]

22 Jul 2022

PONE-D-21-40147

Macroeconomic impact of Ebola outbreaks and potential mitigation of GDP loss with prophylactic Ebola vaccination programs

PLOS ONE

Dear Dr. Morrison,

Thank you for submitting your manuscript to PLOS ONE. After careful consideration, we feel that it has merit but does not fully meet PLOS ONE’s publication criteria as it currently stands. Therefore, we invite you to submit a revised version of the manuscript that addresses the points raised during the review process.

Please note that we have only been able to secure a single reviewer to assess your manuscript. We are issuing a decision on your manuscript at this point to prevent further delays in the evaluation of your manuscript. Please be aware that the editor who handles your revised manuscript might find it necessary to invite additional reviewers to assess this work once the revised manuscript is submitted. However, we will aim to proceed on the basis of this single review if possible. 

We look forward to receiving your revised manuscript.

Kind regards,

Vanessa Carels

Staff Editor

PLOS ONE

Journal Requirements:

2. Thank you for providing the following Funding Statement:  "This work was supported by Johnson and Johnson Services, Inc. The funder did not play a role in the study design, conduct, collection, and management but did provide feedback on data analysis and interpretation, as well as manuscript preparation, review, approval and decision to submit."

We note that one or more of the authors is affiliated with the funding organization, indicating the funder may have had some role in the design, data collection, analysis or preparation of your manuscript for publication; in other words, the funder played an indirect role through the participation of the co-authors. 

If the funding organization did not play a role in the study design, data collection and analysis, decision to publish, or preparation of the manuscript and only provided financial support in the form of authors' salaries and/or research materials, please review your statements relating to the author contributions, and ensure you have specifically and accurately indicated the role(s) that these authors had in your study in the Author Contributions section of the online submission form. Please make any necessary amendments directly within this section of the online submission form.  Please also update your Funding Statement to include the following statement: “The funder provided support in the form of salaries for authors [insert relevant initials], but did not have any additional role in the study design, data collection and analysis, decision to publish, or preparation of the manuscript. The specific roles of these authors are articulated in the ‘author contributions’ section.” 

If the funding organization did have an additional role, please state and explain that role within your Funding Statement. 

Please also provide an updated Competing Interests Statement declaring this commercial affiliation along with any other relevant declarations relating to employment, consultancy, patents, products in development, or marketed products, etc.  

3. Thank you for stating the following in the Competing Interests section: "I have read the journal's policy and the authors of this manuscript have the following competing interests: This research is sponsored by Johnson & Johnson and is related to the development of products licensed to Johnson & Johnson, at which LM, MG, VOM are employed. This work followed a plan to manage any potential conflicts arising from this arrangement, wherein these authors did not contribute to study design, data analysis, or compilation of study results."

We note that you received funding from a commercial source: 'Johnson and Johnson'

Reviewers' comments:

Reviewer's Responses to Questions

**Comments to the Author**

1. Is the manuscript technically sound, and do the data support the conclusions?

Reviewer #1: Partly

2. Has the statistical analysis been performed appropriately and rigorously? 

Reviewer #1: Yes

3. Have the authors made all data underlying the findings in their manuscript fully available?

Reviewer #1: Yes

4. Is the manuscript presented in an intelligible fashion and written in standard English?

Reviewer #1: Yes

5. Review Comments to the Author

Reviewer #1: This is an interesting study that quantifies the potential epidemiologic and economic impact of Ebola vaccination.

Title:

The title and the abstract should mention that the study refers to sub-Saharan African countries.

Abstract:

Include the objectives.

The first sentence of the results is a conclusion.

Include more results in the abstract.

The first two sentences of the conclusion could be deleted.

Include actual conclusions of the study.

Introduction:

Include a table with the main socioeconomic characteristics of the countries evaluated at the start of the Ebola outbreak.

Methods:

The section “Estimating the reduction in cases from Ebola vaccine deployment” includes sets of vaccination effectiveness ratios assumed by the authors and estimated by a previous study (39). Clarify which numbers are used in the model. It could be more appropriate to use the results of the previous study (39). The results of the study in reference 39 include large confidence intervals that could be used for the sensitivity analysis.

There are multiple assumptions related to the effectiveness of the vaccine that are not mentioned in the manuscript.

Table 1 requires more explanation in the text.

It is unclear why the study uses the total number of cases vs. the number of cases per 100,000 population which could be a better parameter for the model given the large differences in population in the countries included in the study. The results of the model using the number of Ebola cases adjusted by population should be included as part of the supplemental materials.

Estimating the impact of Ebola on the GDP per capita in the third year excludes the impact in years 1-2 and after year 3, resulting in an underestimation of the true effect of Ebola on the GDP per capita. This should be mentioned as a limitation of the study.

This type of study should include other countries without Ebola outbreaks in the region to control for changes in GDP trends related to other causes. This should also be mentioned in the limitations section of the study.

Results:

The exclusion of Guinea from the analyses is not justified and it is not mentioned in the methods section.

Table 1. The titles of this table below are confusing and need explanation.

Total (Aggregate) Mitigated GDP Loss (million $)

Share of Actual Impact Mitigated by Vaccine (%)

Remaining GDP per Capita Loss (%)

Discussion:

The first paragraph of the discussion belongs to the results section.

The discussion should include a comparison with previous studies that evaluated the epidemiologic and macroeconomic effects of vaccination for Ebola and other diseases.

Conclusions:

Most conclusions are actually a discussion of the results. Include the main conclusions in this section.

6. PLOS authors have the option to publish the peer review history of their article (what does this mean?). If published, this will include your full peer review and any attached files.

Reviewer #1: **Yes: **Enrique Seoane-Vazquez

---

## [Author Response · Author response to Decision Letter 0]

23 Feb 2023

Dear Editors,

Thank you for these comments and your review; they are very helpful to the improvement of our manuscript, “Macroeconomic impact of Ebola outbreaks in Sub-Saharan Africa and potential mitigation of GDP loss with prophylactic Ebola vaccination programs”, which we have addressed below. We greatly appreciate the time and effort put forth by the reviewer to improve our paper. If any responses are unclear or you wish additional changes to be made, please let us know.

Sincerely, 

Laura Morrison

Senior Economist and Policy Analyst 

RTI International

Reviewer #1 Comments

Title:

1. The title and the abstract should mention that the study refers to sub-Saharan African countries. 

Thank you. We have adopted this suggestion and inserted this clarification into both the manuscript’s title and abstract.

(Lines: 4, 35)

Abstract:

2. Include the objectives.

Agree. We have included the objectives in the abstract, with the goal of orienting the reader to our overall goals initially. 

(Lines: 30-33)

3. The first sentence of the results is a conclusion.

Thank you. We have removed this sentence and merged it with the second sentence for conciseness and to retain a focus on describing the direct results of the work.

(Lines:40-43)

4. Include more results in the abstract.

Agree. We have included several more results in the abstract to better summarize the findings of our work. 

(Lines: 40-46).

5. The first two sentences of the conclusion could be deleted.

Agree. We have deleted the second sentence and condensed the first sentence to be written more like a conclusion per this suggestion. We think it is important to retain the first sentence’s statement around the work’s identification of economic impact, as a conclusion, as it is a key implication of the study.

(Line: 47)

6. Include actual conclusions of the study.

Thank you. We have distilled the content in this section to focus on the main conclusions of our work.

(Lines: 47-49)

Introduction:

7. Include a table with the main socioeconomic characteristics of the countries evaluated at the start of the Ebola outbreak.

Thank you for this suggestion. We have added a table to the methods section of the paper, wherein we introduce the countries included in the study. We included key economic, development, and health indicators that contextualize the socio-economic status of each country relative to each other.

(Lines: 126)

Methods:

8. The section “Estimating the reduction in cases from Ebola vaccine deployment” includes sets of vaccination effectiveness ratios assumed by the authors and estimated by a previous study (39). Clarify which numbers are used in the model. It could be more appropriate to use the results of the previous study (39). The results of the study in reference 39 include large confidence intervals that could be used for the sensitivity analysis.

Thank you for this comment. We address your question regarding the numbers used in the Disease Incidence Calculator and respond to your suggestion around sensitivity analysis below. We have clarified the sources and assumptions in the Table 2 footnotes. 

A 62% and 91% risk reduction are used in the model for low coverage/low efficacy and high coverage/high efficacy scenarios, respectively, per Potluri et al (2020), Technical Report. Input assumptions are drawn from the Potluri modeling analysis results for the 2018 Ebola outbreak in the North-Kivu province of Democratic Republic of the Congo (DRC). The low scenario in our current paper uses the percentage of cases averted from the Potluri base scenario analysis, which assumed 60% vaccine efficacy (i.e., 60% reduced risk of being infected). Our analysis uses the Potluri results that assumed no reduction in infectiousness or case fatality rates from vaccination and vaccination coverage of 30% for HCW and 5% for the general population. Table 14 in Potluri et al. shows an estimated 62% of cases averted under these assumptions. The high scenario in our paper uses the percentage of cases averted from the Potluri et al. upside scenario, which assumed 90% vaccine efficacy (i.e., 90% reduced risk of being infected). Our analysis uses the Potluri results that assumed no reduction in infectiousness or case fatality rates from vaccination and vaccination coverage of 60% for HCW and 10% for the general population. Table 24 in Potluri et al. shows an estimated 91% of cases averted under these assumptions.

Our analysis used the sensitivity analysis findings from the Potluri et al. modeling analysis for the 2018 outbreak in the North-Kivu province of DRC. Specifically, we used the most conservative results from the Potluri baseline and upside scenarios, which assumed that vaccination would reduce the risk of infection among those vaccinated, but not reduce their infectiousness or case fatality rates. The analysis in our paper focused on two plausible scenarios for vaccine efficacy (i.e., reduced risk of infection): 60% and 90%. Our analysis further focused on two plausible scenarios for the percentage of the populations vaccinated: (1) 30% of health care workers (HCW) and 5% of the general population and (2) 60% of HCW and 10% of the general population. It is worth noting that we did not use the Potluri et al. conservative scenario for vaccine efficacy of 30% reduction in the risk of infection because we selected a lower bound vaccine efficacy of 60%. The Potluri et al. sensitivity analysis results for the 30% efficacy scenario indicate that with no reduction in infectiousness or case fatality rate, about 40% of cases would be averted (Table 19). However, the conservative scenario analysis also shows that if vaccine efficacy is only 30%, but vaccination also reduces infectiousness by 50%, then 64% of cases would be averted. This value is close to our lower bound assumption of the percentage of cases averted (62%) and suggests that the assumption of 62% reduction in cases from vaccinating 30% of the high risk and 5% of the general population is a reasonable lower bound. 

(Lines: 240-241)

9. There are multiple assumptions related to the effectiveness of the vaccine that are not mentioned in the manuscript.

Thank you for this suggestion. We have added detail to clarify the table numbers in Potluri et al. that provided our assumed reductions in cases for the low and high scenarios. Specifically, we used the Potluri et al. 2018 DRC sensitivity analysis findings that assume no reduction in infectiousness or case fatality rates among vaccinated individuals. This is the most conservative assumption used by Potluri, as they also examined scenarios that assumed (1) a 50% reduction in case fatality rates only and (2) a 50% reduction in case fatality rates and a 50% reduction in infectiousness. We have clarified the sources and assumptions in the Table 3 (formerly Table 2) footnotes. 

(Lines: 273-274)

10. Table 1 (now Table 2) requires more explanation in the text.

Agree. We have taken this suggestion and included an additional explanatory sentence about how the data in Table 1 (now Table 2) were used in the analysis and what the assumptions indicated for each scenario.

(Lines: 240-241)

11. It is unclear why the study uses the total number of cases vs. the number of cases per 100,000 population which could be a better parameter for the model given the large differences in population in the countries included in the study. The results of the model using the number of Ebola cases adjusted by population should be included as part of the supplemental materials.

Thank you for this suggestion. Using a population-adjusted metric can be useful in some cases, as you suggest, yet there are several key reasons why we chose to use absolute cases (total number of cases) instead. While we state our reasoning for this in our methodology section, beginning on line 129 (as follows), we are providing additional clarification. “Absolute number of cases reported during historical Ebola outbreaks were used as an indicator of the magnitude of the outbreak itself. We used absolute cases instead of population-adjusted cases (e.g., per capita incidence) to avoid understating the risk of Ebola transmission among large populations and the disease dynamics set in motion by the outbreak.”

First, the theoretical reasoning to support this decision, is that using population-adjusted cases masks the considerable economic impact of a small number of Ebola cases in any population, regardless of the size, which we sought to highlight. For instance, when a few cases grows to a few hundred and then to a few thousand, it becomes exponentially more difficult to contain an outbreak. Even a few hundred cases can impact factors which comprise GDP by triggering shocks to trade relationships, causing commercial operations to shut down or scale back, and compelling governments to react strongly with measures that can reduce the spread of the outbreak, but stifle production for the period that these measures are in effect. Because Ebola has a high fatality rate, these reactions typically stem from a small number of cases, such that the absolute number of cases metric is appropriate to use. 

Second, our methodology is set up to initially establish an econometric model of the outbreak that is agnostic to the difference between absolute and population-specific numbers. Using the synthetic control method forms the basis for detecting a causal relationship in this analysis. In our first step, “Estimating causal effects of Ebola outbreaks using synthetic control”), only the time at which the outbreak began matters, which is used to establish the point-in-time estimate at which the outbreak began, and the Ebola-affected country’s GDP is compared to that of non-Ebola affected countries (and a population-adjusted figure would not change the synthetic control estimates.)

Third, in step three of our methodology “The Effect of Ebola on GDP per capita in Ebola-affected countries” we examine the relationship between impact on GDP in the Ebola-affected countries and the size of the outbreaks. This step is what allows us to tie in both the economic impact and the disease incidence data that reflects the impact of a hypothetical Ebola vaccine. This is a purely descriptive task, not an estimation technique, wherein the relationship with a measure of outbreak size (which absolute cases of Ebola is) can be explored, should a meaningful and defensible relationship be identifiable. When plotting this against population-adjusted data (cases per 100,000), the relationship did not demonstrate a clear and compelling relationship, suggesting that cases per 100,000 is not useful for predicting impact on GDP. Absolute cases, however, did show a clear exponential relationship and was thus used as a measure for predicted economic impact. Importantly, this does not change the causal results established in the first step of our methodology (synthetic control). 

We believe that our use of absolute cases is appropriate in this case, in place of a population-adjusted figure in the manuscript, but do appreciate the thoughtful inquiry into this decision in our methodology.

12. Estimating the impact of Ebola on the GDP per capita in the third year excludes the impact in years 1-2 and after year 3, resulting in an underestimation of the true effect of Ebola on the GDP per capita. This should be mentioned as a limitation of the study.

Thank you for this clarifying point: the study does capture the impact of Ebola on GDP in aggregate over years one, two, and three following the outbreak, and is presented as such in Table 4 (previously Table 3). This is what we mean when we say impact “by year 3”, which requires further clarification. While we do discuss the quantitative impact occurring in year 3 in the results text, the values that are presented most frequently in tables and text are aggregated GDP loss aggregated over three years. By calling this to our attention, we have been able to make an adjustment to the written content in several sections to clarify this point.

(Lines: 180-181, 344-355, 358, 363)

13. This type of study should include other countries without Ebola outbreaks in the region to control for changes in GDP trends related to other causes. This should also be mentioned in the limitations section of the study.

We agree that this it is important to have a reliable counterfactual comprised of like countries that do not experience Ebola outbreaks. This study’s design and the synthetic control methodology it employs does use other regionally co-located countries that did not experience Ebola outbreaks as a counterfactual, which is the “synthetic control” against which an Ebola-affected country’s observed GDP outcomes are compared. The synthetic control is formed using a weighted combination of comparison units determined by an algorithm that tries to match the pre-outbreak country’s characteristics as best as possible. This is explained in detail beginning on line 159 in our methodology. For this reason, we have elected not to make any additions to the limitations section of our manuscript.

Results:

14. The exclusion of Guinea from the analyses is not justified and it is not mentioned in the methods section.

Thank you for this valuable suggestion. We have added a sentence to clarify why Guinea is not included in subsequent steps following the synthetic control estimation in the methodology.

(Lines: 256-260)

15. Table 1. (Table 4) The titles of this table below are confusing and need explanation.

Total (Aggregate) Mitigated GDP Loss (million $)

Share of Actual Impact Mitigated by Vaccine (%)

Remaining GDP per Capita Loss (%)

Thank you for pointing out this lack of clarity in table titles. Table 4 (formerly Table 3) has these column headers and we have both added additional language to the table and we have sought to explain further in the notes beneath the table (lines 341-346). The new column headers read as follows (line 337):

- Total GDP Loss Mitigated by Vaccine (million $, aggregate GDP loss over three years)

- Share of GDP Loss from Outbreak Mitigated by Vaccine (%)

- Remaining GDP Loss (%)

(Lines: 358, 362-367)

Discussion:

16. The first paragraph of the discussion belongs to the results section.

Agree. The majority of this paragraph was moved to the results section, while a portion that explores implications of this work for investment in preventative measures, like vaccination, was retained in the discussion section.

(Line: 382-389)

17. The discussion should include a comparison with previous studies that evaluated the epidemiologic and macroeconomic effects of vaccination for Ebola and other diseases.

Thank you for this suggestion. We have added a sentence in the discussion that compares our estimates with other studies that have assessed comparable impacts of Ebola (most notably those examining the 2014 West Africa outbreak.) While these studies do consider a different, more limited set of factors, they exhibit a remarkably wide range of estimates, which our results fall within.

(Lines: 406-409) 

Conclusions:

18. Most conclusions are actually a discussion of the results. Include the main conclusions in this section.

Agree. We have removed the content that did not speak directly to conclusions and retained only the central conclusion of the work. 

(Lines: 490)

---

## [Editor Report · Decision Letter 1]

15 Mar 2023

Macroeconomic impact of Ebola outbreaks in Sub-Saharan Africa and potential mitigation of GDP loss with prophylactic Ebola vaccination programs

PONE-D-21-40147R1

Dear Dr. Laura T.R. Morrison,

We’re pleased to inform you that your manuscript has been judged scientifically suitable for publication and will be formally accepted for publication once it meets all outstanding technical requirements.

Kind regards,

Ricky Chee Jiun Chia

Academic Editor

PLOS ONE
---

## [Editor Report · Acceptance letter]

31 Mar 2023

PONE-D-21-40147R1 

Macroeconomic impact of Ebola outbreaks in Sub-Saharan Africa and potential mitigation of GDP loss with prophylactic Ebola vaccination programs 

Dear Dr. Morrison:

I'm pleased to inform you that your manuscript has been deemed suitable for publication in PLOS ONE. Congratulations! Your manuscript is now with our production department. 

Kind regards, 

on behalf of

Dr. Ricky Chee Jiun Chia 

Academic Editor

PLOS ONE